# Triplet Test on Rubble Stone Masonry: Numerical Assessment of the Shear Mechanical Parameters

**Michele Angiolilli**  **and Amedeo Gregori \***

Department of Civil, Building and Environmental Engineering, University of L'Aquila, 67100 L'Aquila, Italy; michele.angiolilli@graduate.univaq.it

\* Correspondence: amedeo.gregori@univaq.it; Tel.: +39-0862434141

**Abstract:** Rubble stone masonry walls are widely diffused in most of the cultural and architectural heritage of historical cities. The mechanical response of such material is rather complicated to predict due to its composite nature. Vertical compression tests, diagonal compression tests, and shear-compression tests are usually adopted to investigate experimentally the mechanical properties of stone masonries. However, further tests are needed for the safety assessment of these ancient structures. Since the relation between normal and shear stresses plays a major role in the shear behavior of masonry joints, governing the failure mode, a triplet test configuration is herein investigated. First, the experimental tests carried out at the laboratory of the University of L'Aquila on stone masonry specimens are presented. Then, the triplet test is simulated by using the total strain crack model, which reflects all the ultimate states of quasi-brittle material such as cracking, crushing, and shear failure. The goal of the numerical investigation is to evaluate the shear mechanical parameters of the masonry sample, including strength, dilatancy, normal, and shear deformations. Furthermore, the effect of (i) confinement pressure and (ii) bond behavior at the sample-plate interfaces are investigated, showing that they can strongly influence the mechanical response of the walls.

**Keywords:** unreinforced masonry; quasi-brittle material; in-plane behavior; shear-compression; triplet test; dilatancy; bond behavior; confinement; finite element model; macro-model

## 1. Introduction

In the past, the traditional architecture of Mediterranean countries extensively used stone, especially limestone, giving rise to one of the most important parts of the historical heritage, despite the serious damage inflicted by recurring earthquakes [1–5].

The extreme vulnerability of stone masonry buildings is mainly due to the mortar joints, which represent the weak zone in masonry systems [6]. The weakness of the mortar joints is particularly relevant in the case of strong units combined with weak mortar joints, which is the typical condition in the case of ancient stone masonries [7].

Although in the last few decades, several laboratory and in situ tests were performed on stone masonry walls, the mechanical behavior of stone masonries is still not completely characterized due to a lack of experimental data [8]. Indeed, due to the composite nature of the ancient masonry structures, the high irregularity of elements, and the complex distributions of mortar joints, the mechanical response of irregular stone walls is difficult to reproduce both in experimental tests and in numerical simulations. Here, it is worth highlighting the great effort made in some recent studies [9,10] to correlate the quality of the masonry walls with their mechanical properties. It is common practice to employ vertical compression [11–14], diagonal compression [8,11,15–17], and shear-compression [18,19] tests to investigate experimentally the mechanical properties of stone masonries. However, further tests are needed to obtain useful data for safety assessment studies of ancient masonry structures [8].

Several experimental studies have been carried out on the bond shear strength of unit–mortar interfaces [20,21] and natural rock joints [7,22,23], but limited research is available on the shear behavior of stone masonry joints. The knowledge gathered can be partly extended to the present study knowing that the surface roughness plays an important role in the shear behavior of stone masonries.

One can find few experimental campaigns carried out by using the triplet test for rubble stone masonries (see [8,24]), in which reliable experimental evaluation of shear parameters is difficult to perform since results may be scattered because the mortar joints are not regularly arranged. However, the triplet test may be considered as a valid alternative to the other destructive tests due to the smaller size of the specimen required. The smaller specimen size, the easier operation, and the lower costs (and the invasiveness in the case of existing buildings) are relevant. Indeed, the size of the specimen tested under the triplet test can be assumed equal to about 50 mm in length and 50 mm in height (see [8,24]) and is more than two times smaller than the other tests. Thus, the triplet test may overcome the limitations of the other destructive tests that do not allow an extensive characterization.

The goal of the study carried out in the present work was to investigate the most important mechanical parameters governing the shear behavior of the traditional rubble stone masonry walls. In particular, first, the paper presents results from triplet tests carried out at the laboratory LPMS (Laboratorio di Prove Materiali e Strutture) of the University of L'Aquila (Italy). Then, numerical simulations are presented to better understand the mechanical behavior of stone masonry under the triplet test, also investigating the effect of the confinement pressure and the bond behavior at the masonry-plate interface. The conclusions of the present study offer the possibility to improve the experimental mechanical characterization of stone masonry structures and the modeling of them.

## 2. The Triplet Test

### 2.1. Description of the Test

The triplet test allows the evaluation of the shear strength parameters of the bed joints of the masonry. This test can be effective to generate shear failure, through the mortar of the masonry specimen, especially in the case of strong units combined with weak mortar joints, which represent the most frequent condition in ancient stone masonries. Indeed, for this masonry type, the mechanism failure mainly regards the mortar joints, which represent the weak zone in masonry systems [6,7]. The triplet test relies on a particular constraint system that creates a shear box [24]. Usually, a couple of rigid steel plates is mounted on the upper and lower parts of the wall specimen (see Figure 1a). Then, a combined application of vertical pressure on the upper plate and a horizontal force on the unconstrained lateral edge of the specimen causes the slide of the central zone of the sample on two horizontal surfaces (see Figure 1b).

The testing setup is an important issue to comprehend the experimental characterization of the shear behavior of the mortar joints of masonry walls [25]. Indeed, as highlighted in [26], the results are sensitive to the support conditions used. Although distinct loading arrangements have been used, it is difficult to provide uniform shear and normal stresses distribution along the joint so that failure occurs simultaneously at all points of the mortar joint [27]. That condition can be obtained by reducing the eccentricity of the reactions that may develop as close as possible to the unit-mortar interfaces [25]. The standard shear test method, also called the triplet test (EN1052-3 [28]), provides the best testing setup for the evaluation of the shear parameters of masonry walls. However, that test was considered for brick masonry walls. Unfortunately, no standards exist about mechanical tests on masonry wall samples made of irregular stone elements distributed in a chaotic texture. For stone masonry, the triplet test is more difficult to perform because the rocking phenomenon caused by the irregularity of the stone arrangement may preclude the shear sliding.

In the absence of specific standards for the testing of irregular stone masonry walls, the test configuration employed here referred to the standard code (EN1052-3 [28]) used for brick masonries. In particular, the shear stress of the specimen $f_v$ is obtained from the equation:

$$f_v = H/(2A) \tag{1}$$

where $H$ is the shear load and $A$ is the cross-sectional area of the shear surface equal to 0.12 m$^2$ (0.40 × 0.30 m). Since $H$ was applied to the central part of the specimen and transferred to the upper and lower parts through the two contact surfaces, the computation of $f_v$ considers the total area of shear surfaces equal to $2A$ (EN1052-3, [28]). For $H = H_{max}$, Equation (1) provides the maximum shear stress.

When moderate normal stress is applied on the masonry panel, the friction resistance assumes the most important role for the shear characterization of the masonries also due to the negligible resistance of the mortar [21,29]. In that condition, the Coulomb criterion can be assumed for the evaluation of the shear strength of masonry walls, accurately describing only their local failure [21,30,31], by the equation:

$$f_v = f_{v0} + \mu\,\sigma \tag{2}$$

where $f_{v0}$ is the cohesion, which represents the shear stress at zero vertical load stress, and $\mu$ is the friction coefficient. Reliable evaluation of the $f_{v0}$ and $\mu$ parameters is difficult to perform in the case of stone masonry because they strongly depend on the asperity of the stones and may not be considered representative for the entire masonry structure [32]. For stone units coupled with hydraulic lime mortar, $f_{v0}$ values ranging from about 0.08 to 0.3 were experimentally measured [8,33]. In national [34] and international codes [35], $\mu$ is considered as constant, independent of the wall type, and assumed equal to 0.4. Instead, experimental values of such coefficient were measured, in the case of stone units coupled with hydraulic lime mortar, ranging from about 0.3 to 1.2 [8,33,36].

By combining Equations (1) and (2), one can characterize the shear mechanical properties of the masonry specimens.

An important aspect regarding shear tests is the dilatancy $\psi$, which represents the relation between the normal and the shear displacements of the wall ($\psi = \arctan \Delta_n/\Delta_s$) and assumes a significant role in numerical modeling of rubble stone masonry [25]. Indeed, an increase in the volume of stone specimens can be observed during the test because; after cracking and sliding, the two sides of the cracks do not match. For this reason, the dilatancy of stone masonry is mostly controlled by the joint roughness, as already observed for rock joints [37]. Indeed, the dilatancy increases with the irregularity of the crack surfaces and tends to be stronger in rubble stone masonry than in brick masonry [8]. As pointed out by [38], dilatancy in masonry panels causes a significant increase in the shear strength when they are subjected to confinement loads.

### 2.2. Description of the Experiments

To quantify the shear strength parameters of horizontal bed joints in rubble stone masonry, triplet tests were performed at the laboratory LPMS of the University of L'Aquila on two stone masonry samples measuring about 0.50 m in length, 0.50 m in height, and 0.30 m in thickness. Samples were prepared using the original limestone units and the ancient constructive technique recognized in most of the monumental buildings of L'Aquila. Irregular stone elements of a calcareous nature and with a characteristic size ranging from 80 mm to 150 mm were taken from the debris of buildings collapsed under the L'Aquila 2009 earthquake. The original mortar features of the historical masonry (characterized by a very friable behavior and a low compressive strength of about 2 MPa) were reproduced in several attempts. In particular, the mortar was prepared by mixing commercial natural hydraulic lime mortars, local crushed limestone sand, and local natural clay with a respective ratio of 1:2:1. Water was added to the mortar mixture until a plastic consistency was reached. The addition of natural clay to the mortar mixture produced a hydraulic lime similar to the ancient lime [39].

Wall samples were consolidated by mortar injections to preserve their integrity during the delicate movement operations.

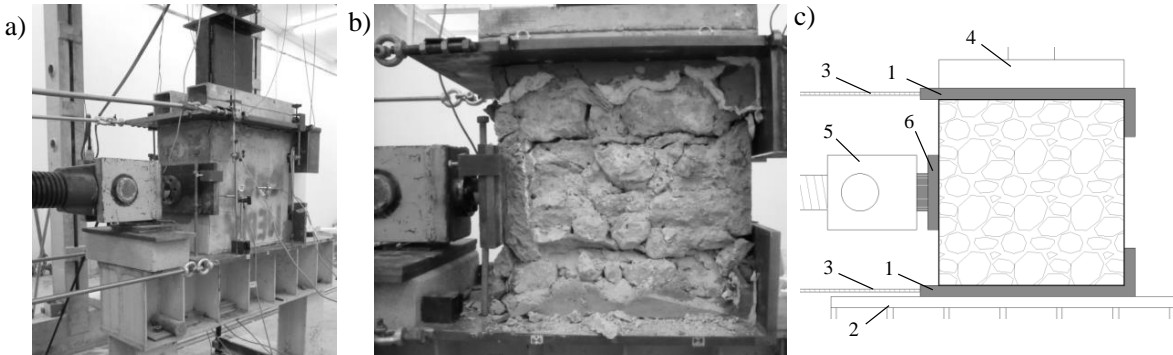

**Figure 1.** (**a**) Testing apparatus employed for the triplet tests carried out at the laboratory LPMS (Laboratorio di Prove Materiali e Strutture) of the University of L'Aquila. (**b**) Specimen failure. (**c**) Experimental details of the test.

Here, it is relevant to say that the test equipment consisted of a pair of rigid steel plates (measuring 30 mm in thickness) mounted around the masonry wall panels to simulate a shear box (see Item 1 in Figure 1c). The bottom plate was placed on a rigid steel basement (see Item 2 in Figure 1c). Both the top and bottom steel plates were anchored by steel bars (see Item 3 in Figure 1c), which did not allow horizontal translations of the plates.

As far as the load system was concerned, first, a vertical actuator was used to impose an axial load on the sample (see Item 4 in Figure 1c), reaching the average compression stress of −0.18 MPa (equal to about 1/3 of the failure stress experimentally obtained from the compression of a sample), which was kept constant for all the duration of the tests. Second, a horizontal actuator (see Item 5 in Figure 1c) was used to introduce a shear force $H$ at the half-height of the sample by using a lateral steel plate (see Item 6 in Figure 1c).

Due to the boundary and loading conditions designed in the experiments, the central zone of the sample was forced to slide on the upper and lower specimen parts through two distinct shear surfaces.

Further details about the manufacture of the masonry samples, the test equipment, and the test execution were provided in [24].

Figure 1b shows the crack propagation occurring during the experimental test. In particular, one can clearly observe a crack concentration at the horizontal layers between the lateral plate and the two L-shaped plates. Lower amount cracks also occurred at the bottom and upper parts of the specimens. It is worth noting that a rotation of the upper part of the specimen was observed during the tests.

The mechanical response of the specimens carried out by experimental tests is described in the Section 4.

## 3. Description of the Numerical Model

Extensive research on advanced numerical modeling and analysis of historic masonry structures has been carried out for some decades [40–42]. However, reliable prediction of the mechanical response of such a material is still a challenge for engineers [43].

Different modeling approaches are available for the numerical simulation of the mechanical behavior of masonry structures. Actually, the equivalent frame models [44,45], the macro-models, also called continuous models [46–50], and micro-modeling, also called discontinuous models [51–53], may be adopted based on the different details to which the material heterogeneity is required to be represented. In particular, the discontinuous approach can give better results, especially when the geometry is known, but is computationally expensive for the analysis of large masonry structures since

failure zones are placed in pre-assigned weak zones, such as the mortar joints for brick masonries [54]. Instead, the continuum approach performs well in the case of damage zones spread over the wall [55].

Previous researches showed that the response of masonry structures up to failure can be successfully modeled using techniques applied to concrete mechanics because both are characterized by brittle behavior [56–58].

In the present work, the in-plane compressive behavior of UnReinforced Masonry (URM) walls was investigated by using a macro-modeling approach, where the heterogeneous material was substituted with an equivalent homogeneous material. In particular, the experimental tests described in Section 2.2 were simulated by using Midas FEA [59]. This FE code can be used for simulating the behavior of quasi-brittle materials, such as stone masonries, by employing the Total Strain Crack Model(TSCM) [60]. The TSCM is often used for macro-modeling of masonry [47–49]. It is based on the modified compression field theory originally proposed in the multi-direction fixed crack model [61] and extended to 3D by [62]. The model is based on a smeared crack approach, where the process of cracking is obtained by "smearing" the damage on the adjacent finite element, introducing a degradation of the relevant mechanical properties [63]. The model also offers a variety of possibilities to consider the orientation of the crack, ranging from fixed single to fixed multi-directional and rotating crack approaches [64,65]. Since smeared crack modeling approaches do not require remeshing of the FE model after the occurrence of cracks or the a priori definition of possible locations of cracks, they have been widely used in FE modeling [66]. The smeared crack models are practice-oriented due to the limited data required in the input and, for example, are successfully adopted for brick masonry and adobe walls [67,68] and debonding problems [66,69,70].

In the present work, under the TSCM hypotheses, the fixed stress-strain concept was used, so that the axes of crack remained unchanged once the crack was activated. Furthermore, both the lateral crack and the confinement effect were considered. The system of non-linear equations used a secant stiffness matrix and was solved by the initial stiffness incremental method.

The shear-stress constitutive laws employed for the numerical simulation of stone masonries in [63,66] were adopted in this study. In particular, as assumed for the compressive law, also the tensile law included a softening branch, thought to be important for the non-linear analyses. Indeed, even if a simplified brittle law can be adopted for the characterization of the tensile behavior, this choice may not always be appropriate. In particular, the compression behavior of the masonry was modeled by the constitutive law suggested in [71], characterized by a parabolic hardening path and a parabolic exponential softening branch after the peak of resistance (Figure 2a). The tensile behavior was instead modeled by Thorenfeldt's law [72] characterized by a linear hardening branch followed by a nonlinear softening branch (Figure 2b).

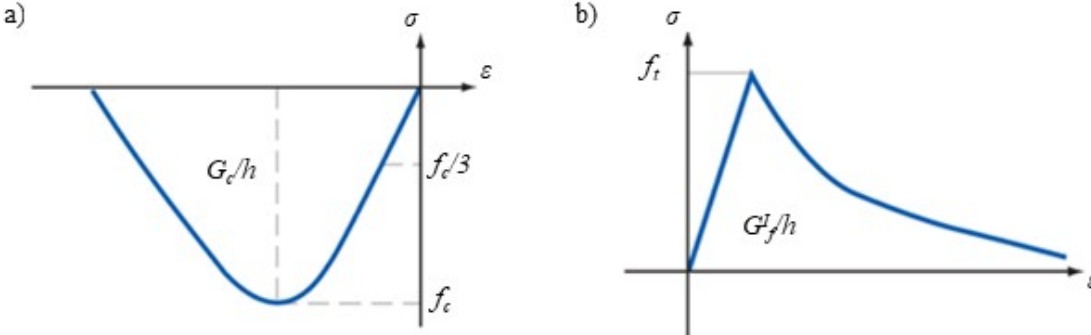

**Figure 2.** Stress-strain constitutive relations: (**a**) Masonry uniaxial compression. (**b**) Masonry uniaxial tension

The stone masonry panel was modeled by the FE macro-model M1 illustrated in Figure 3a. The M1 consisted of a single block of homogeneous material characterized by a hexahedral mesh with a size of 50 mm. In a continuum modeling approach, the mesh size should be larger than the

size of the aggregates and other dominant micro-structure features. However, due to convergence issues of the analyses, it was decided to adopt here a mesh size slightly smaller than the average stone size. This choice ensured higher reliability of the numerical results and no strong differences in the mechanical response of the masonry specimen, as confirmed by previous sensitivity analyses, not reported in this paper.

The two L-shaped plates, placed at the lower and the upper parts of the specimen (see Items 1 and 2 in Figure 3a), were modeled as tetrahedral elements and were assumed to be elastic ($E$ = 210,000 MPa). The same assumption was adopted also for the lateral plate (see Item 3 in Figure 3a) used to transfer the horizontal displacement to the central zone of the sample.

As far as the load was concerned, two different loading steps were applied to the panel aiming to reproduce the same condition of the experimental tests: (i) a constant vertical compression stress of $-0.18$ MPa on the upper L-shaped plate (see Item 2 in Figure 3a) to have a uniform distribution of the vertical load and (ii) a horizontal displacement, gradually increasing from 0 to 20 mm, transferred by the lateral plate to have the sliding of the central zone of the sample.

As far as the constraint system was concerned, the vertical translations together with the rotations perpendicular to the normal plate directions were fixed for all the bottom mesh nodes. Furthermore, only horizontal translations were fixed for the outermost mesh points of the two L-shaped plates. The top of the sample was left free to move in the vertical direction like in the experimental tests. That constraint system allowed reproducing the same boundary of the experimental tests. Details of the constraint system adopted in the simulation are illustrated in Figure 3b.

Since the small number of iterations and steps used for the numerical analyses may affect the quality of the calculated responses, a huge number of iterations and steps were assumed for the analyses. In particular, 150 iterations and 200 steps, which corresponded to an increment of 0.1 mm per step, were assumed to solve the nonlinear equation system by using the incremental Newton–Raphson method.

## 4. Simulating the Experimental Tests

### 4.1. Calibration of the Material Parameters

The composite nature of masonry makes it difficult to assign material properties, which depend on many factors as described above. For this reason, the first important part of the work consisted of calibrating the mechanical parameters in order to reproduce the mechanical behavior observed during experimental tests.

The masonry mechanical properties required by the model were the normal elasticity modulus $E$, the shear modulus $G$, the compressive strength $f_c$, the tensile strength $f_t$, and the compressive and tensile energy fractures, $G_{f_c}$ and $G_f$, respectively. The values of $f_c$ and $f_t$ were assumed equal to those ones obtained experimentally in [24]. The normal elasticity modulus $E$ and the shear modulus $G$ were defined by the best fitting of both the $H$–$d_x$ and $f_v$–$\varepsilon_v$ experimental curves recorded during the tests performed in [24]. Such a fitting procedure was carried out by sensitivity analyses, not reported in the paper. On the other hand, it was necessary to calibrate the values for $G_{f_c}$ and $G_f$ to model the inelastic behavior of the URM. In particular, the $G_{f_c}/h$ and $G_f/h$ ratios represent the the area under the stress-strain ($\sigma-\varepsilon$) diagrams of Figure 2, where $h$ represents the crack bandwidth, and it can be assumed equal to the average mesh size adopted in the FE model [59,73]. However, any experimental investigation presented in the literature allowed a reliable characterization of both the $G_{f_c}$ and $G_{fI}$ for stone masonries. Empirical formulations were used for the estimation of such parameters:

$$G_{f_c} = 15 + 0.43\, f_c - 0.0036\, f_c^2 \tag{3}$$

$$G_{fI} = 0.025\, (2\, f_t)^{0.7} \tag{4}$$

One computes a value of 15.2 N/mm for $G_{f_c}$ by using Equation (3) and a value of 0.0075 N/mm for $G_{f_c}$ by using Equation (4). Such values were computed with the $f_c$ and $f_t$ values listed in Table 1. However, it is worth noting that such equations were proposed for brick masonries [74]

by modifying the formulation originally proposed for concrete material [75]. In the case of stone masonries, Equation (3) tends to overestimate the $G_{f_c}$ value. Indeed, for example, the value assumed for $G_{f_c}$ in the numerical analyses carried out for the Camponeschi Palace [66], which is a historical stone masonry building located in L'Aquila, was assumed equal to 9 N/mm. Values of 3 N/mm were instead estimated for $G_{f_c}$ in the case of tuff masonry material [63,76,77].

On the contrary, the $G_{fI}$ value obtained by Equation (4) slightly underestimated the values presented in the literature [54,66]. Sensitivity analyses on such mechanical parameters would deserve a deeper investigation and should be considered in future studies.

Table 1 shows the masonry parameters obtained by the described calibration procedure. It should be noted that the ratio $E/G = 2.2$ was close to the value provided by the Italian Building Code [34] for irregular stone masonries assumed equal to three.

**Table 1.** Mechanical parameters calibrated for the masonry panel under shear-compression.

| $E$ (MPa) | $G$ (MPa) | $f_c$ (MPa) | $G_{fc}$ (N/mm) | $f_t$ (MPa) | $G_{fI}$ (N/mm) |
|-----------|-----------|-------------|-----------------|-------------|-----------------|
| 100 | 45 | 0.7 | 15.2 | 0.09 | 0.0075 |

Figure 3c shows the best fitting of the experimental curves. In particular, the grey area in the graph represents the scattering area between the lower and the upper response curves measured in the two experimental tests, whereas the black curve concerns the numerical simulation.

By focusing on the experimental responses Figure 3c, one can see first an almost linear branch, a subsequent softening behavior, and a final plastic horizontal tail. The two specimens reached the maximum shear force ranging from about 70 kN and 80 kN, showing some small differences in stiffness. That difference may be related to friction generated on the masonry-plate interfaces. After the peak strength was attained, both specimens showed a plastic behavior related to the sliding of the central part of the specimen through the two shear surfaces. The horizontal force recorded from the specimens remained constant even for large displacement values.

The numerical curve illustrated in Figure 3c, obtained by an accurate choice of the parameter values and the fitting procedure described above, well described the mechanical behavior of the experimental tests. In the rest of the paper, the numerical curve of Figure 3c is taken as the reference curve for the $H$–$dx$ plots.

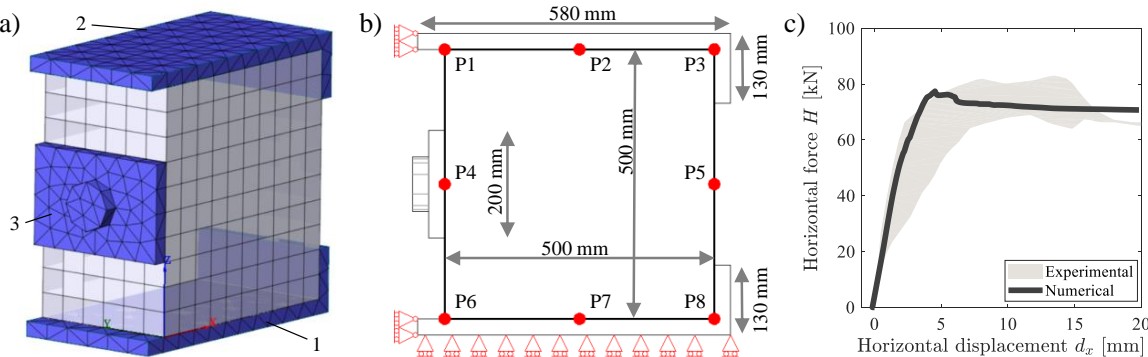

**Figure 3.** (**a**) Visualization of the M1 macro-model. (**b**) Scheme adopted in the test. (**c**) Best fitting of the $H$–$d_x$ experimental response.

No extensive literature reports regard the triplet test on stone masonry specimens since the test is commonly employed only for brick masonries. However, the results of both the experiment and simulation presented in this paper were in line with the literature results obtained for the same masonry type and same materials [8,78]. Here, the maximum shear stress considered in this study ($f_v = 0.26$ MPa) was found in good agreement with the values computed in [8,78] ($f_v$ ranging from

0.33 MPa to 0.44 MPa and obtained for a similar vertical compression stress). The slight difference among all these results corresponded to the natural scattering observed in terms of the mechanical response from stone masonry walls (as described in Section 1).

### 4.2. Numerical Assessment of the Shear Mechanic Parameters

The experimental tests carried out in [24] measured the relation $H$–$d_x$ (Figure 3c) up to the failure of the URM sample, whereas the relation $f_v$–$\varepsilon_v$ was evaluated only in the elastic phase (up to 1/3 of the shear strength). However, to better investigate the mechanical behavior of the URM specimen under the triplet test, it was important to measure also other parameters. In this work, the displacement of eight points (from P1 to P8) placed on the face of the panel (see Figure 3b) was monitored during the simulation aiming to compute the shear displacement $\Delta_s$, the normal displacement $\Delta_n$, the horizontal strain $\varepsilon_h$, and the vertical strain $\varepsilon_v$. In particular, $\varepsilon_h$ and $\varepsilon_v$ were computed for both the middle and lateral parts (left, right, top, and bottom) of the sample, as follows:

$$\varepsilon_{h,middle} = -(d_{x,P4} - dx_{P5})/\ell \tag{5}$$

$$\varepsilon_{h,top} = -(d_{x,P1} - dx_{P3})/\ell \tag{6}$$

$$\varepsilon_{h,bottom} = -(d_{x,P6} - dx_{P8})/\ell \tag{7}$$

$$\varepsilon_{v,middle} = (d_{z,P2} - dz_{P7})/\ell \tag{8}$$

$$\varepsilon_{v,right} = (d_{z,P3} - dz_{P8})/\ell \tag{9}$$

$$\varepsilon_{v,left} = (d_{z,P1} - dz_{P6})/\ell \tag{10}$$

$$\Delta_s = d_{x,P5} - (d_{x,P3} + d_{x,P8})/2 \tag{11}$$

$$\Delta_n = d_{z,P3} \tag{12}$$

where $\ell$ is the length between two points, equal to 500 mm, and $d_x$ and $d_z$ are the horizontal and the vertical displacements of the eight points placed on the face of the panel.

Results of the numerical simulations are illustrated in Figure 4a–f. Plots indicate the achievement of the maximum shear stress $f_{v,MAX}$ by the "x" marker and called the MSP label (Maximum Stress Point) in the following. Instead, when the achievement of the maximum normal and shear displacements occur at the same point, $\Delta_{n,MAX} \equiv \Delta_{s,MAX}$, the "o" marker and the MDP label (Maximum Displacement Point) are used in the paper.

In detail, Figure 4a shows the relation between $f_v$ and the horizontal strains $\varepsilon_h$ computed for the three parts of the sample: middle, top, and bottom. In particular, one can see that the strain values computed for the top and the bottom parts of the specimen assumed a constant value almost equal to zero because of the nearness of the L-shaped plates that constrained the horizontal displacement of the specimen. For the middle part of the sample, an initial, almost linear branch was first recognized up to the MSP, then the mechanical response of the masonry was characterized by a gradual reduction in $f_v$ up to MDP, after which a large increase in $\varepsilon_h$ was observed for a constant residual value of $f_v$. In the rest of the paper, the numerical curve of the middle part of the sample represented in Figure 4a is taken as the reference curve for the $f_v$–$\varepsilon_h$ plots.

Figure 4b shows the vertical strains $\varepsilon_v$ computed for three parts of the sample: middle, right, and left. In particular, for all three curves, one can observe compressive values (negative) equal to $-0.002$ MPa for $f_v = 0$ MPa. This was due to the compressive load applied to the upper L-shaped plate in the first loading step. Then, one can see a gradual increase in $\varepsilon_v$ for all three curves up to the MSP, after which a large increase in $\varepsilon_v$ was observed for a constant residual value of $f_v$ up to the MDP. After the achievement of the MDP, one can observe that $f_v$ continued to decrease because of the damage of the specimen, which reduced the capacity to transfer the force $H$ (and the related $f_v$) to the masonry specimen. After the achievement of the MDP, one can also observe a decrease in the $\varepsilon_v$ value

because the damage on the specimen reduced its vertical load-carrying capacity. As a consequence, the initial vertical expansion of the specimen decreased as a result of the larger axial deformation under the constant vertical compression stress $\sigma$.

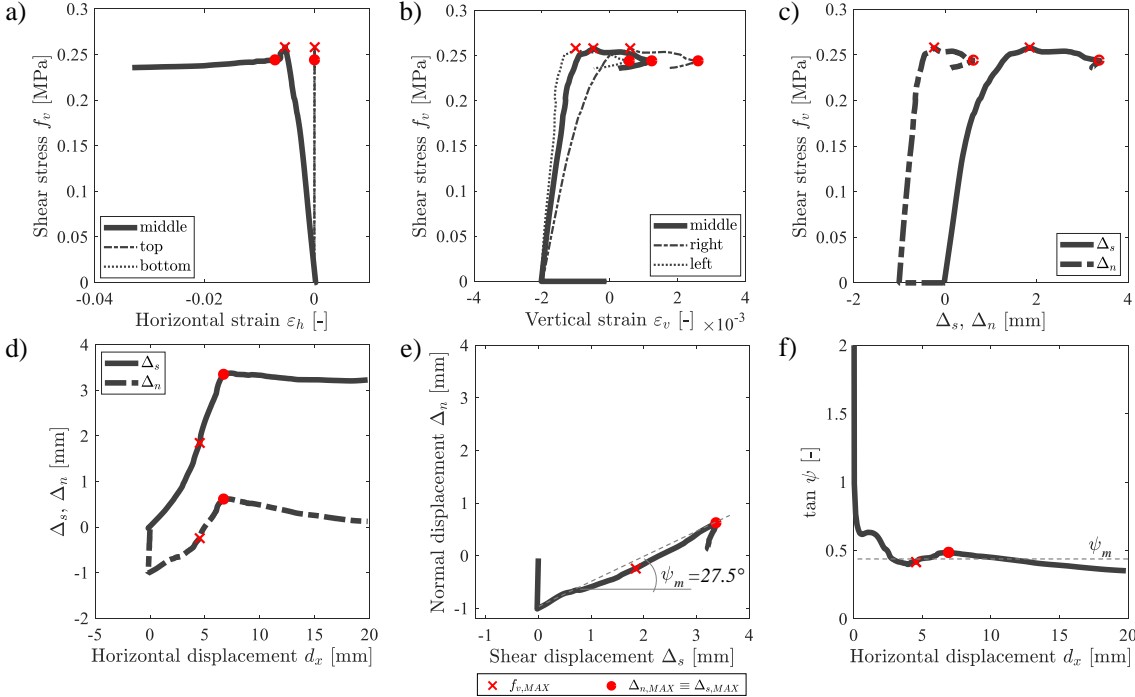

**Figure 4.** Shear stress expressed in terms of: (**a**) horizontal strain, (**b**) vertical strain (**c**) normal and shear displacements. (**d**) Normal and shear displacements function of $d_x$. (**e**) Graphical estimation of the mean dilatancy angle $\psi_m$. (**f**) Variation of $\tan \psi$ during the application of $d_x$.

Furthermore, analyzing the differences between the three curves of Figure 4b, one can see that the right and left parts of the specimen were characterized by different values of $\varepsilon_v$ as compared to the middle part. This was due to the rotation of the specimen that also occurred in the experimental test.

Figure 4c shows the correlation between shear stress $f_v$ and both normal and shear displacements, $\Delta_n$ and $\Delta_s$. In general, the trends and the comments of Figure 4b,c are transferable for Figure 4c.

Figure 4d shows the variation of both $\Delta_n$ and $\Delta_s$ as a function of the displacement of the lateral plate $d_x$. In particular, one can observe an initial increase of both $\Delta_s n$ and $\Delta_s$, which continued to increase also after the achievement of the MSP. After the achievement of the MSP, one can observe a decrease in the $\Delta_n$, which meant that a relaxation phenomenon of the masonry material was occurring. Moreover, one can observe that $\Delta_n$ slightly dropped to a residual value maintained almost constant even at large displacements. This was due to the damage concentration on the masonry portion close to the lateral plate. From Figure 4d, it is clear that the trend variation in $\Delta_n$ and $\Delta_s$ after the achievement of the MDP was obtained gradually during the test; this information is difficult to obtain by only observing Figure 4c.

As discussed in Section 2.1, it is known from the literature that quasi-brittle materials show an increase in volume when undergoing inelastic shear deformations. This phenomenon depends on both the confinement pressure and the dilatancy angle $\psi$. The dilatancy $\psi$ represents the relation between the vertical and the horizontal displacement of the wall ($\psi = \arctan \Delta_n / \Delta_s$). Figure 4e shows the relation between $\Delta_s$ and $\Delta_n$ computed for the specimen, allowing graphically estimating the mean dilatancy angle, equal to 27.5°. That curve is taken as the reference curve for the $\Delta_n$–$\Delta_s$ plots in the rest of the paper. To better investigate the variation of the dilatancy angle during the test, Figure 4f shows the relation between $\tan \psi$ and the displacement of the lateral plate $d_x$. One can observe that the $\tan \psi$ assumed a high value during the compressive loading phase ($d_x = 0$) and an almost constant value

equal to 0.48 during the application of the lateral displacement $d_x$. The results were in line with the results of Van der Pluijm [79], who experimentally established values of tan $\psi$, ranging from 0.2 to 0.7 for low confinement pressures, highlighting the strong influence of the confinement on the estimation of $\psi$.

To better understand the variation of both the displacement and stress fields during the triplet test, Figure 5a,b shows the plots corresponding to incremental values of the displacement $d_x$ imposed at the lateral plate. In particular, the plots of Figures 5a,b are referred to $d_x$ equal to 1 mm, 4.5 mm, 6.9 mm, and 20 mm, ensuring that one observed the mechanical behavior at the elastic phase (pre-peak), maximum shear stress, maximum shear and normal displacements, and the end of the test. The displacement field is illustrated in Figure 5a, showing that the masonry part close to the lateral plate tended to be subjected to larger deformation since the perfect bond behavior at the plate-masonry interfaces reduced the deformation capacity of the other part of the specimen. The stress field (von Mises plots) is instead illustrated in Figure 5b, showing a stress concentration on the upper and lower corners of the masonry.

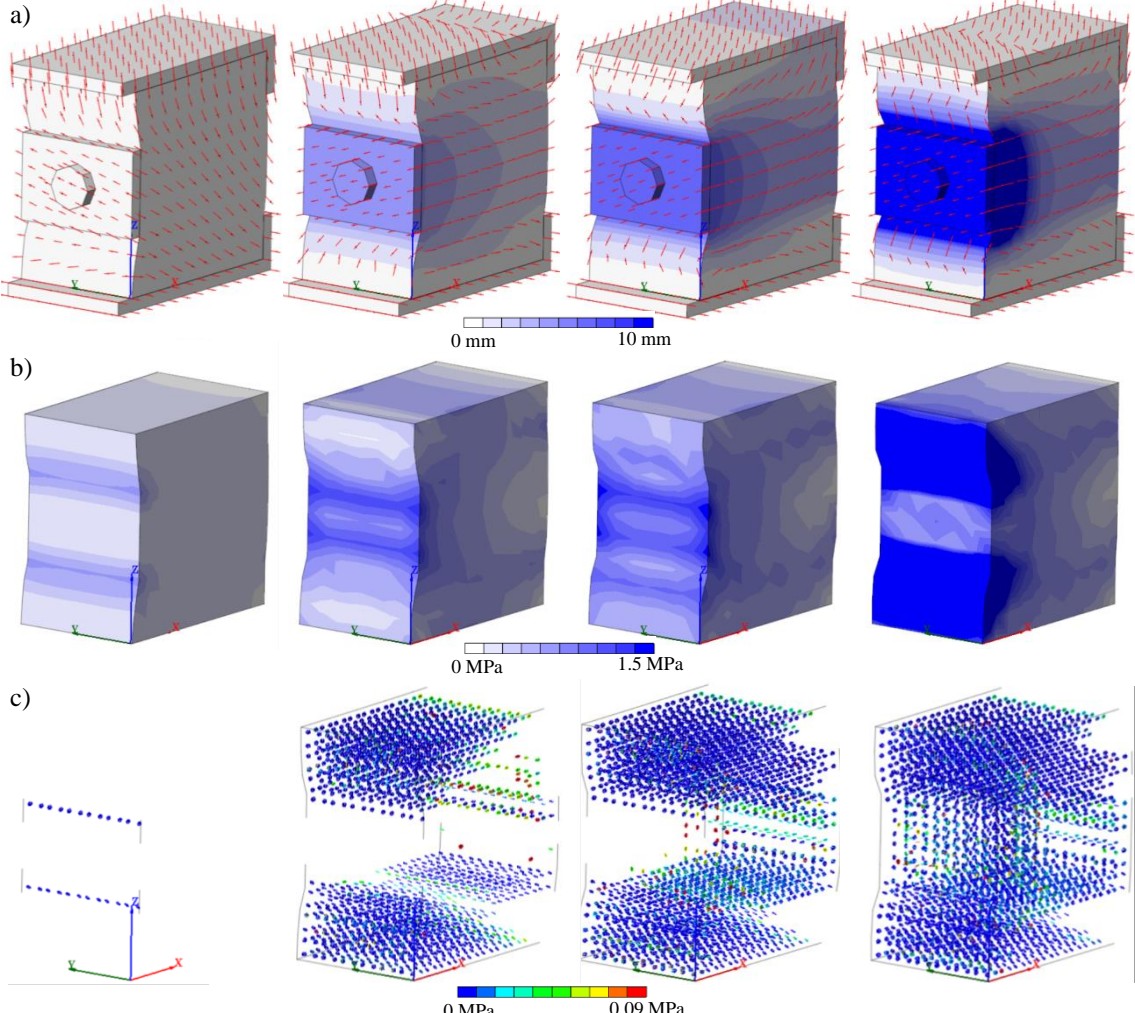

**Figure 5.** Masonry specimen for incremental values of the displacement $d_x$ corresponding to the elastic phase (1 mm), maximum shear stress (4.5 mm), maximum shear and normal displacement (6.9 mm), and the end of the test (20 mm), illustrated in terms of: (**a**) Displacement field (Dxyz plots). (**b**) Stress field (von Mises plots). (**c**) Evolution of the cracking pattern.

Furthermore, Figure 5c shows the evolution of the cracking pattern of the masonry sample for incremental values of $d_x$ in terms of Gauss point-occurrence, indicating whether a tension cut-off limit

was exceeded at an integration point. In particular, when the stress limit was exceeded at a given integration point, this point was marked with a color in the model view, with the indication of the actual stress reached in that position. After the occurrence of the crack, the stress of the integration point tended to decrease, more or less quickly based on the softening behavior used for the material to characterize the tensile behavior, up to arriving at 0 MPa, when that Gauss point did not contribute anymore to the material resistance. By observing Figure 5c, one can see that the upper and lower parts of the specimen were characterized by larger damage level as compared to the central part of the specimen. In detail, crack concentration occurred at the horizontal layer between the lateral plate and the two L-shaped plates and the corner places. The failure on the two corners was due to the perfect bond behavior between the masonry and the two "L-shaped" plates and the huge difference in the elasticity modulus *E* of their material.

Definitely, the cracking pattern observed in Figure 5c highlights that the simplified numerical modeling, which considered all nonlinear behavior of the masonry sample concentrated on the sliding surfaces, while keeping the three parts of the specimen as elastic, may produce incorrect results.

### 4.2.1. Confinement Effect

The confinement effect, which can take place between two contact surfaces, was considered in the analyses by assuming the Selby–Vecchio law [62] in the material properties, as discussed in Section 2.1. Results presented in the following (Figures 6 and 7) describe the variation of the mechanical properties of the masonry sample by varying the value of the confinement pressure, assumed equal to $-0.18$ MPa ($\sigma_0$ case), $-0.09$ MPa ($0.5\,\sigma_0$ case), and $-0.36$ MPa ($2\,\sigma_0$ case).

The results of Figure 6 highlight that the confinement level affected the strength of the specimen. Moreover, for the case of low confinement pressure ($0.5\,\sigma_0$ case), one can observe a hardening behavior of the response, whereas for the other two cases, a softening behavior is observed. Furthermore, it is worth noting that, for the case of $0.5\,\sigma_0$, the shear displacement and the normal displacement occurred at the same value of $d_x$, as already observed for the $\sigma_0$ case. On the contrary, for the case of $2\,\sigma_0$, one can observe that $\Delta_{n,MAX}$ and $\Delta_{s,MAX}$ occurred in two different phases. In particular, in the $2\,\sigma_0$ case, $\Delta_{n,MAX}$ and $\Delta_{n,MAX}$ occurred before and after the achievement of the MSP, respectively.

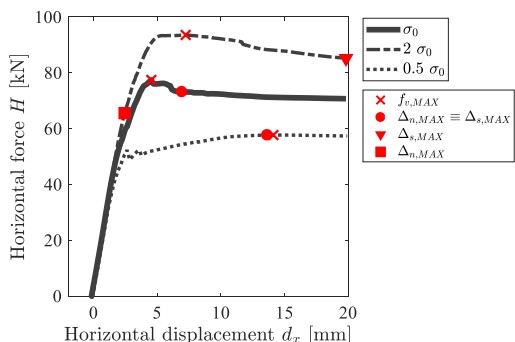

**Figure 6.** Confinement effect in the horizontal displacement-horizontal force plot.

Figure 7a shows the relation between the normal displacement $\Delta_n$ and the shear stress $f_v$. It is worth noting that no considerable lifting of the masonry sample could be observed in the case of high confinement pressure ($2\,\sigma_0$ case). Indeed, after the initial compressive phase that led to the normal displacement equal to $-2$ mm, a slight lifting of the sample was observed up to $f_v = 0.22$ MPa, for which the maximum value of $\Delta_n$ was computed. After that point, the sample continued to be compressed, showing a decrease in $\Delta_n$. This trend highlighted that high confinement pressure limited the lifting of the specimen. For the case of low confinement pressure ($0.5\,\sigma_0$ case), one can see that the MSP coincided with the MDP, and they occurred at the end of the test. The maximum value in $\Delta_n$

showed that that the specimen under low confinement pressure was subjected to a considerable lifting of the specimen that could be difficult to manage in the experimental test.

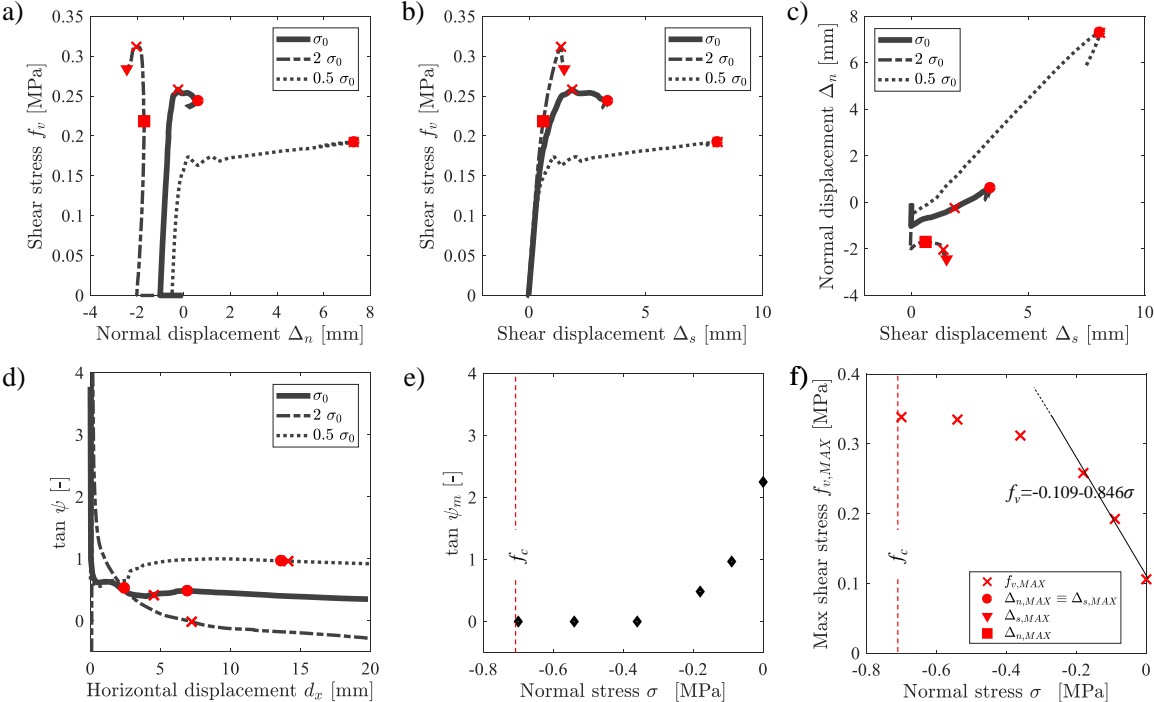

**Figure 7.** Shear mechanical properties of the masonry sample by varying the confinement pressure with respect to the central value $\sigma_0 = -0.18$ MPa: (**a**) $f_v - \Delta_n$ plot. (**b**) $f_v - \Delta_s$ plot. (**c**) $\Delta_n - \Delta_s$ plot. (**d**) $\tan \psi - d_s$ plot. Effect of the confinement pressure $\sigma$ in terms of: (**e**) $\tan \psi_m$. (**f**) $f_{v,MAX}$.

Figure 7b shows the relation between the shear displacement $\Delta_s$ and the shear stress $f_v$. For all three curves, the achievement of the maximum value of the shear displacement $\Delta_{s,MAX}$ was observed after the achievement of the MSP. Moreover, $\Delta_{s,MAX}$ occurred at the end of the test only for the case of high confinement pressure, as already clearly observed in Figure 6.

Figure 7c shows the relation between $\Delta_s$ and $\Delta_n$ computed for the specimen allowing graphically estimating the mean dilatancy angle $\psi_m$. Values of $\tan \psi$ are analyzed in Figure 7d as a function of the lateral displacement $d_x$. In that figure, one can see that, after the initial phase ($d_x$ almost equal to zero), the dilatancy tended to be constant. Moreover, the higher the confinement pressure, the higher the $\tan \psi$ value was, with a null value of $\tan \psi$ in the $2\sigma_0$ case.

Additional analyses (Figure 7d,e) were carried out also assuming values of $\sigma$ equal to 0 MPa, $-0.54$ MPa, and $-0.7$ MPa to better understand the variation of both $\tan \psi$ and $f_{v,MAX}$ as a function of the confinement pressure. In particular, the results of Figure 7d show that $\tan \psi$ was ranging from zero to 2.25, following an exponential law. Figure 7f allows defining the limit strength domain of masonry and estimating the values of the cohesion $f_{v0}$ and the coefficient of friction $\mu$. In particular, $f_{v0}$ and $\mu$ were computed equal to $-0.109$ MPa and $-0.846$ MPa, respectively, by assuming a linear regression of the results obtained for $\sigma_0$ equal to 0 MPa, $-0.18$ MPa, and $-0.36$ MPa.

### 4.2.2. Bond Effect at the Masonry-Plate Interface

The analyses carried out in Sections 4.2 and 4.2.1 regarded the hypothesis of a Perfect Bond (PB) behavior at the specimen-plate interfaces. However, in the experimental tests, the masonry specimens could move along the horizontal direction, whereas only the horizontal displacement of the L-shaped plates was constrained. Here, for a sliding at the specimen-plate interface, a Weak Bond (WB) hypothesis was assumed in the FE model. In particular, contact plane elements were introduced by assuming the Coulomb friction nonlinear law [59] with values of 0.05 MPa for the cohesion $c$, 30° for

the internal friction angle $\phi$, and 80 N/mm$^3$ and 35 N/mm$^3$ for the normal and the tangential stiffness $k_n$ and $k_t$, respectively.

The results presented in the following (Figure 8) describe the variation of the mechanical properties of the masonry sample by varying the bond behavior at the sample-plate interfaces.

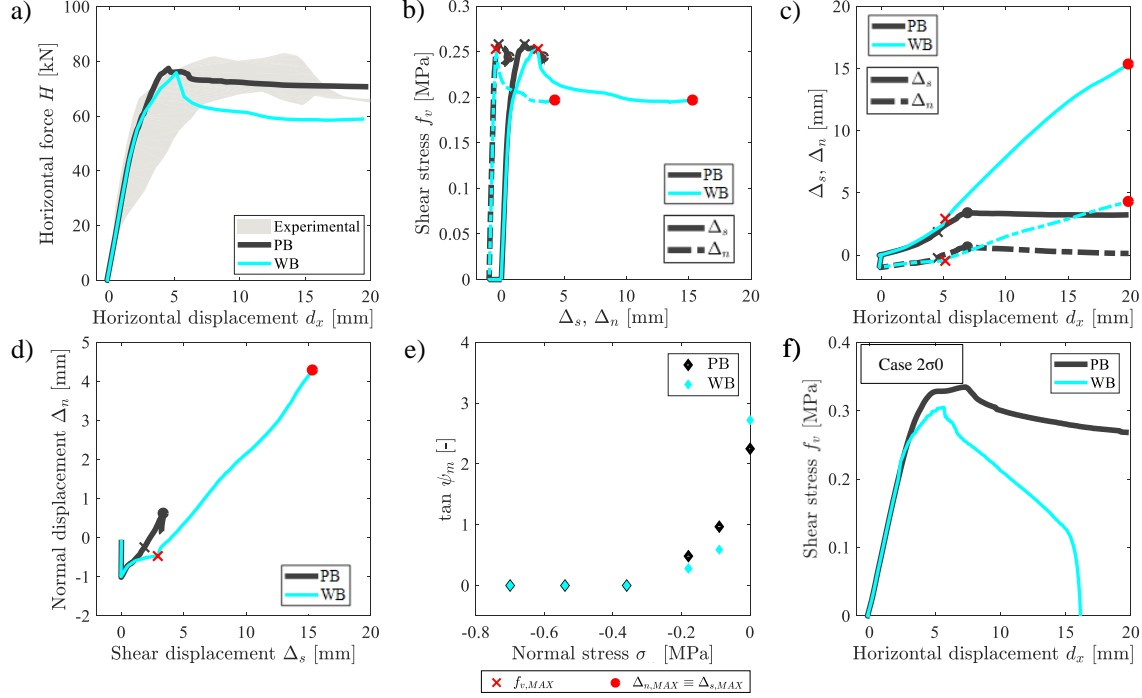

**Figure 8.** Numerical results obtained for two bond behavior hypotheses at the masonry-plate interfaces: Perfect Bond (PB) and Weak Bond (WB). (**a**) $H - d_x$ plot. (**b**) $f_v - \Delta_s$ and $f_v - \Delta_n$ plots. (**c**) $\Delta_s - d_x$ and $\Delta_n - d_x$ plots. (**d**) $\Delta_n - \Delta_s$ plot. (**e**) $\tan \psi_m - \sigma$ plot. (**f**) $f_v - d_x$ plot for the $2\sigma_0$ case.

Figure 8a shows that the bond behavior had no effect on the first branch of the $H - d_x$ plot and had a slight effect on the maximum shear load (about 2%). On the contrary, a larger difference in terms of load-bearing capacity for large displacement $d_x$ was observed between the WB and the PB cases. Indeed, by measuring $H$ at $d_x$ = 20 mm, one can compute a load decrease of 17 % by comparing the PB (71 kN) and WB (59 kN) cases. Moreover, after the achievement of about 60 kN of the horizontal force $H$, one can observe a higher decrease in the slope of the response curve of the WB case, as compared to the PB case. This is related to the sliding of the masonry on the two "L-shaped" plates through the two interfaces introduced in the WB case. It is worth noting that the experimental response showed the same trend in the initial phase of the $h - d_x$ curve, even more emphasized. On the contrary, one can see that the load-carrying capacity observed for large displacements of the PB case was similar to the experimental case, as compared to the WB case. This difference was related to the higher damage on the lateral portion of the specimen (where the displacement was applied) in the WB case, as compared to both the PB case and the experimental case.

Figure 8b shows the relation between both the normal and shear displacements with the shear stress $f_v$. In particular, strong differences could be observed in terms of the MDP value, which was higher in the WB case as compared to the PB case. This meant that the boundary condition highly influenced the deformation capacity of the masonry specimen.

Figure 8c shows the variation of both $\Delta_n$ and $\Delta_n$ as a function of the lateral displacement $d_x$. The plot clearly highlights the higher deformation capacity of the masonry sample in the WB case as compared to the PB case. Indeed, for the PB case, one can see that $\Delta_n$ and $\Delta_n$ assumed constant values after a certain value of $d_x$ (almost equal to 6 mm), whereas a gradual increasing in both $\Delta_n$ and $\Delta_n$ up to the end of the test was observed for the WB case.

Figure 8d shows the relation between $\Delta_s$ and $\Delta_n$ computed for the specimen. In particular, one can see that the dilatancy angle was higher for the PB case as compared to the WB case.

The results of Figure 8e show the particular variation of the trend of tan $\psi_m$ function of the normal stress $\sigma$ by assuming PB and WB hypotheses.

A further analysis was carried out to investigate the effect of the boundary condition on the mechanical response of the masonry subjected to high confinement pressure. Figure 8f shows the $f_v - d_x$ plot for high confinement pressure ($2\,\sigma_0$ case). In particular, one can compute a decrease in the shear strength of about 11% by comparing the PB case (0.350 MPa) and the WB case (0.312 MPa). Moreover, one can note a total decay of the shear stress for the WB case, whereas a good bearing-load can be observed for the PB case, which was characterized by a decrease in $f_v$ of about 20% as compared to the peak value. Definitively, by comparing the response of Figure 8a,f, one can say that the higher the confinement pressure, the higher the influence of bond behavior on the mechanical response.

Finally, Figure 9 shows the effect of the bond behavior at the masonry-plate interfaces on the displacement and stress fields and the fracture propagation in the case of high confinement pressure ($2\,\sigma_0$).

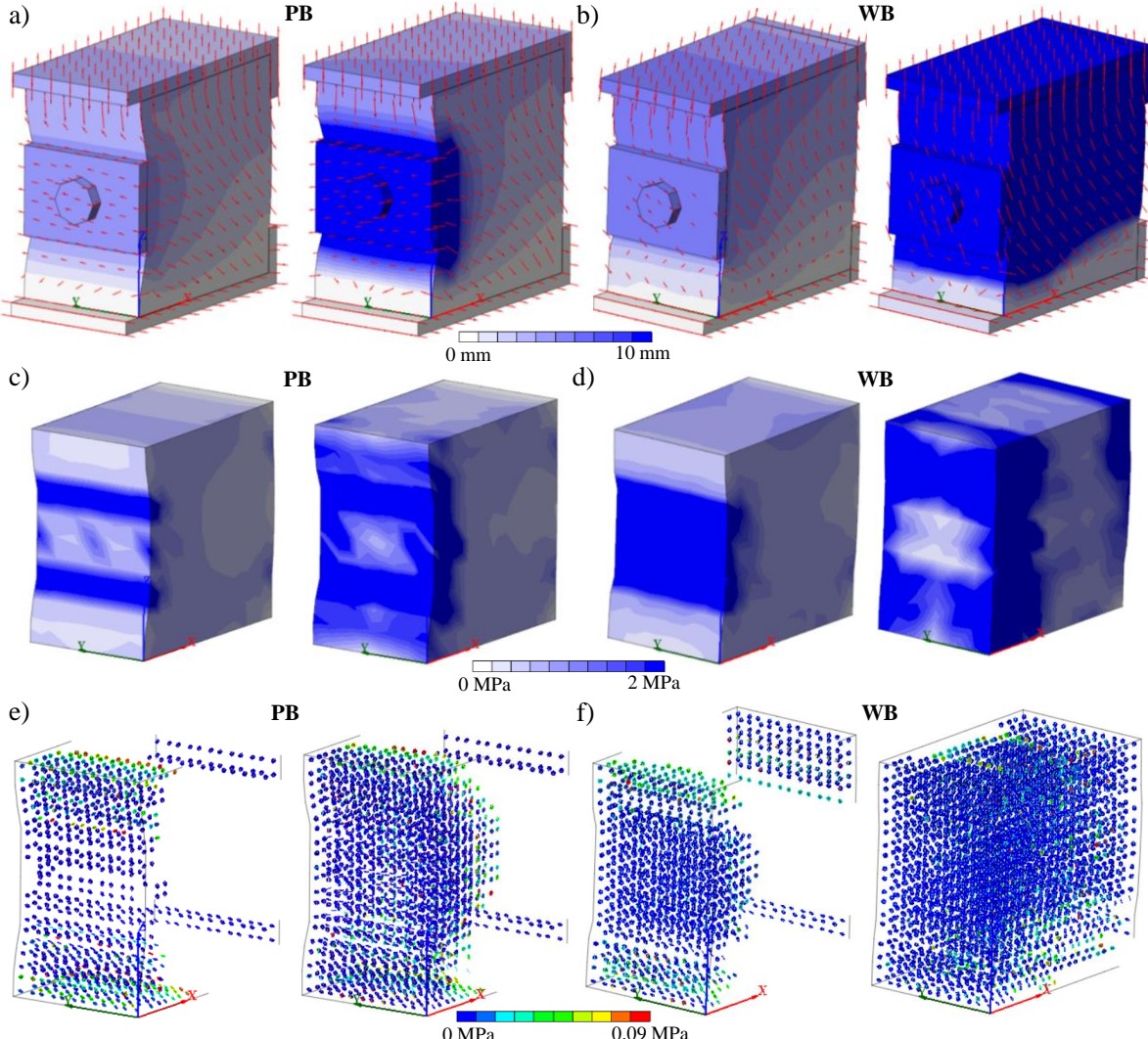

**Figure 9.** Results obtained for high confinement pressure ($2\,\sigma_0$ case) and plotted for two different steps ($d_x$ = 4.5 mm and $d_x$ = 16 mm) by varying the bond behavior at the masonry-plate interfaces (Perfect Bond (PB) and Weak Bond (WB) hypotheses). Imagines are illustrated in terms of: (**a**,**b**) Displacement field (Dxyz plots). (**c**,**d**) Stress field (von Mises plots). (**e**,**f**) Evolution of the cracking pattern.

The plots concern different values of the displacement $d_x$, equal to 4 mm and 16 mm. In particular, one can see clear differences in the sample deformation by assuming the PB hypothesis (Figure 9a) and the WB hypothesis (Figure 9b) because the two L-shaped steel plates of the PB case highly limited the deformation capacity of the specimen. Differences in terms of the von Mises stress for the PB and WB hypotheses are investigated in Figure 9c,d. As far as the fracture propagation was concerned, first, one can see that the WB case (Figure 9e) led to a higher cracking amount as compared to the PB case (Figure 9f). Second, it is worth noting that one can observe different cracking patterns by assuming different confinement pressure and the same bond behavior hypothesis. Indeed, under the PB hypothesis, the damage pattern on the sample for the 2 $\sigma_0$ case (Figure 9e) concerned its lateral parts, whereas one can see cracks' concentration at the lower and bottom parts of the sample for the $\sigma_0$ case (Figure 5c).

Definitely, even if the PB case led to a load-carrying capacity of the specimen similar to the experimental case, the trend of the initial part of the $H$–$d_x$ curve and the propagation failure on the masonry wall observed for the WB case could be considered most realistic for the simulation of the experimental tests, with respect to the PB case.

## 5. Conclusions

The numerical assessment of the shear mechanical parameters on unreinforced stone masonry walls under the triplet test configuration was carried out in the present work.

First, the paper introduced experimental tests carried out at the laboratory LPMS of the University of L'Aquila on several stone masonry samples prepared by using the original limestone units and the ancient constructive technique recognized in most of the monumental buildings of L'Aquila.

Second, the tests were numerically simulated by using a macro-model under the total strain crack assumption, in order to determine the dilatancy, displacement, deformation, and strength of the sample. The results showed a good agreement between experiments and numerical simulations in both the mechanical behavior and the damage evolution on the masonry sample. Then, the effect of the confinement pressure and the bond behavior at the masonry-plate interfaces was also investigated.

In particular, as far as the confinement pressure was concerned, simulations showed that it highly affected the shear strength and the dilatancy of the sample. In addition, it was found that the shear and normal maximum displacements did not occur at the same time when a high level of the confinement pressure was applied, unlike the lower confinement pressure cases.

As far as the bond behavior was concerned, simulations showed that it had a strong effect on the load-bearing capacity for large lateral displacements imposed at the masonry sample. Moreover, the bond behavior had a slight effect in terms of the maximum shear stress and also affected the beginning of the non-linear phase of the wall material. These effects were even more evident by increasing the confinement pressure on the masonry sample. Even if the perfect bond case led to a load carrying capacity of the specimen similar to the experimental case, the stiffness of the mechanical response and the propagation failure on the masonry wall observed for the weak bond case could be considered most realistic for the simulation of experimental tests, with respect to the perfect bond case.

The damage evolution on the unreinforced stone masonry sample depended on both the confinement pressure and the bond behavior, as highlighted by observing the evolution of the cracking pattern. In the case of a moderate level of confinement pressure, possible failure could occur at the upper and lower parts of the sample, also concentrated on the horizontal layer between the lateral plate and the two L-shaped plates. Instead, in the case of high confinement pressure, the cracking mainly occurred at the lateral part of the sample (close to the lateral plate used to apply the shear force to the sample). Finally, the weak bond behavior led to a more uniform cracking as compared to the perfect bond behavior case.

Definitely, the results obtained in this paper highlighted that a simplified numerical model, which considered all nonlinear behavior of the masonry sample concentrated on the sliding surfaces (while keeping the three parts of the specimen as elastic), may produce incorrect results in the numerical

evaluation of the mechanical response of the masonry specimen under the triplet test. Indeed, the cracks on the masonry panel, which strongly depended on both the boundary condition and the confinement pressure, could be propagated also in other parts on the masonry sample.

**Author Contributions:** M.A., A.G. contributed equally to this paper. All authors read and agreed to the published version of the manuscript.

**Acknowledgments:** The Ph.D. scholarship of the first author was co-financed by the Project "2014-2020 PON" (CCI 2014EN16M2OP005).

**Conflicts of Interest:** The authors declare no conflict of interest. The funders had no role in the design of the study; in the collection, analyses, or interpretation of data; in the writing of the manuscript; nor in the decision to publish the results.

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
