# Peer review of "Triplet Test on Rubble Stone Masonry: Numerical Assessment of the Shear Mechanical Parameters"

_buildings, doi:10.3390/buildings10030049_

Round 1

Reviewer 1 Report

This manuscript investigates the triplet test to understand the failure mode of masonry joints because of the relationship between normal and shear stresses.  The experimental tests were carried out on stone masonry specimens, whereas the numerical simulation was done by using the total strain crack model.  The authors may want to address the following comments before consideration for publication.

1. Please elaborate on the experimental details of the triplet test for the readers who are not familiar with this concept.

2. Please provide the boundary and loading conditions for both the experiment and simulation.

3. Please justify the choice of parameters used in this study (e.g., the TSCM, stress-strain relations in Fig. 2).

4. Why is there a gray envelope of experimental responses and how is it obtained?

5. Please benchmark the results from the test/simulation of this study against those in the literature reports to directly demonstrate the novelty of this work.

6. Please provide the physical insights into the results to help clearly explain the underlying physics of the results. 

7. The manuscript would benefit from professional editing (e.g., missing author information).

Reviewer 2 Report

Manuscript: Triplet test on rubble stone masonry: numerical assessment of the shear mechanical parameters.

The authors present experimental tests and numerical simulation for triplet tests of rubble stone masonry. The paper is well written and the results are interesting. Following comments are given for the authors to further consideration:

Abstract: “proprieties” is a mistyped.

Page 1, Line 27: Introduction: steel -> still

Page 1, Line 32-33: “Is common practice employ…” -> the sentence is not clear. Is that a question or a statement sentence?

Page 3, Line 112: “A is the cross-sectional area of the two shear surfaces.” -> total area of shear sufaces should be “2A”

Figure 1(a) was not mentioned in the text.

How many tests (Figure 1b) were repeated?

Page 5, Line 173: “an horizontal” -> it should be “a horizontal”

Page 5, Line 183: corresponds -> correspond

Page 6, Line 203-204: “One compute values…” -> the sentence is not clear

Page 6, Line 210: slight underestimates -> slightly underestimates

Page 7, Line 236: three part -> three parts

Figures 4(b-c): why there is a decrease in both epsilon_v and f_v after achieving MDP?

Page 9, Line 282: consider -> considers

Page 12, Line 349: “Figure 8f shows the H - dx plot for…” -> Figure 8f doesn’t show the H - dx plot but it shows the f_v – dx plot. What is the unit of vertical abscissa, MPa or Pa? Why does the shear stress with WB drop to zero in Figure 8f, considering that horizontal force H (in Figure 8a) is not diminished?

Page 13, L387-388: Conclusions: “…sample highlights that may vary…” => missing a subject in the sentence.

In section 2, there is only one subsection 2.1. Then I think it can be merged in Section 2 because there no more subsections. Similar comment is for Section 3.

Reviewer 3 Report

This paper reports on a combined experimental and numerical
study of the shear deformation and failure of stone masonry
specimens.  The experiments invoke the triplet test on
limestone-based samples.  The modeling uses finite element
method (FEM) with the total strain crack model for inelasticity
and fracture. The effects of confining pressure and bond
behavior at sample plate interfaces was also studied with FEM.

This paper provides new data and modeling results that improves
our understanding of the deformation and failure of masonry
materials.  The methods and results seem credible and should
be of interest to the community and the journal readership.
Some issues should be addressed in revision, however:

[1] Some more information on the composition and microstructure
of the samples should be included.  For example, the components
of the mortar and stone should be given precisely, preferably
in a table.  Also, the size of the aggregate and other
microstructure features should be stated.

[2] For the continuum modeling approach to be valid, the
size of the aggregate and other dominant microstructure
features should be smaller than the element size.  Some
comments on this should be included.

[3] On p. 3, why is f_v = H_max/2A rather than f_v = H_max/A?
Why is the factor of 2 needed on the shear stress?  Please
explain/correct as needed.

[4] This reader did not understand Fig. 5(c).  These do
not look like cracks.  And why are the units for cracks
in MPa?  Please add more explanation of what this figure
on the cracking pattern means.

[5] Similar comment for the bottom part
of Fig. 9: the physical meaning of these images with
the dots is unclear.  Also, Fig. 9 has parts a) through
f) but the figure caption only goes from a) to c).

[6] Different bonding behaviors and other model features
are studied numerically, but it is unclear which
model features are most realistic compared to the
test observations and data.  Please state
more strongly the best modeling approach to be
used so others can follow this recommendation.

[7] The reason for the blurred/wide curve for the
experiment in Fig. 3(c) should be explained.  Is
this intended to show the uncertainty or error
bars in the data?

[8] Paper has some typos.  For example, line
27, "steel", line 58 "most issues", line 299
"oh high".

Round 2

Reviewer 1 Report

In addition to the responses, please include a separate section of "our modifications to the manuscript" for each question from the reviewer.  It is also helpful to highlight the changes made to the manuscript.

Round 3

Reviewer 1 Report

The authors have addressed the comments.